# Academic business research: Impact on academics versus impact on practice

**Vivek Astvansh**[1,2,3,4]*, **Ethan Fridmanski**[5]

**1** Marketing Area, Desautels Faculty of Management, McGill University, Montréal, Québec, Canada,
**2** Department of Informatics, Luddy School of Informatics, Computing and Engineering, Indiana University,
Bloomington, Indiana, United States of America, **3** Environmental Resilience Institute, Indiana University,
Bloomington, Indiana, United States of America, **4** Dewey Data Inc., Solana Beach, CA, United States of
America, **5** Herman B. Wells Library, Indiana University, Bloomington, Indiana, United States of America

* Vivek.astvansh@mcgill.ca

Academic business research: Impact on academics
versus impact on practice. PLoS ONE 18(12):
e0289034. https://doi.org/10.1371/journal.
pone.0289034

Technology, POLAND

**Data Availability Statement:** We use data from
Web of Science and Altmetric.com. We are not
allowed to deposit these data to the journal but
users can obtain the Web of Science data from
their university and request data from Altmetric.

## Abstract

Business journalists and editors of academic business journals have lamented that academic research has little use for any nonacademic stakeholders, including companies, nonprofits, regulators, and governments. Although emotionally unsettling, these commentaries are bereft of evidence on how well a journal's academic impact (measured by impact factor) translates into practice impact. The authors provide this evidence. Specifically, they sample 56 journals, spanning 12 business disciplines, from 2000 to 2020. For each journal-year, they measure two- and five-year *impact factor*, which proxies the impact on academics. Next, for each article published in each journal-year, they collect *attention score*—a weighted sum of the number of times the article is cited in 19 types of practitioner outlets—from Altmetric. The authors then measure the correlation coefficient between the impact factor and attention score for each journal in periods of two-year and five-year. The coefficient indicates how well the journal's academic impact has translated into practice impact. Among the 12 disciplines, international business discipline tops the chart, while information systems, accounting, and finance occupy the bottom positions. *American Economic Review* leads the 56 journals, with *Journal of Marketing Research* and *California Management Review* as close followers. The findings highlight the impact of academic business research —or the lack thereof.

## Introduction

Umpteen journalists (e.g., [1–3]) and editors of academic journals (e.g., [4–8]) have lamented that academic business research has "no obvious value to people who actually work in the world of business" ([2], p. 2). This paradox is consequential to the business academy it is expected to produce knowledge that is useful for a broad set of business stakeholders, including companies, lawmakers, regulators, lawyers, nongovernmental and nonprofit organizations, and society. However, the irony that although so many people have commented on the academy-practice divide, *no one* has quantified the divide. We address this irony in the hope that it reignites solutions to the paradox.

**Funding:** The author(s) received no specific funding for this work.

**Competing interests:** The authors have declared that no competing interests exist.

Specifically, we sample 56 academic journals belonging to 12 business disciplines. Next, we follow a three-step procedure. First, we use the Web of Science Journal Impact Factor (IF) database to compute two-year and five-year academic IF, which is the metric that academic journals use to report their impact on academics. Second, we source from Altmetric—a company that monitors the impact on academic research on nonacademics—their Altmetric Attention score on each article published by each of these 56 academic journals from 2000 to 2020 ([9, 10]). The Altmetric Attention score measures an article's citations in news, social media, public policy documents, patents, blogs, etc. and thus is the most comprehensive measure of the impact of academic research on practice. We use these scores to compute for each journal-year the two-year and five-year attention score. Third, for each article and each discipline, we compute the Pearson product-moment correlation coefficient between (1) the two-year IF and two-year attention score, and (2) the five-year IF and the five-year attention score.

We hypothesize substantial variation in the correlation coefficients across the 12 disciplines. Specifically, we expect the empirically inclined disciplines—such as accounting, finance, information systems, and marketing—to have higher coefficients than their theoretically leaning peers, such as general management and international business. Somewhere in between these two extremes would lie disciplines that are a mix of theory and empirics. These disciplines include economics, operations, and statistics.

We also expect variation across journals within a discipline. This variation would be explained by whether a journal emphasizes managerially actionable insights or contributions to theory. For example, the marketing discipline has four journals in the University of Texas Dallas' list of 24 journals. "*Journal of Marketing* (*JM*) develops and disseminates knowledge about real-world marketing questions useful to scholars, educators, managers, policy makers, consumers, and other societal stakeholders" ([11], p. 1). In contrast, *Marketing Science* is the "premier journal focusing on empirical and theoretical quantitative research in marketing" ([12], p. 1). We thus expect *JM*'s correlation coefficient (between IF and attention score) to be higher than that of *Marketing Science*.

The correlation coefficients report three insights. First, international business (IB) discipline—via its flagship *Journal of International Business Studies* (*JIBS*)—is the most effective in matching academic impact with practice impact. Entrepreneurship is a close second discipline. On the other end of the spectrum are information systems (IS), accounting, and finance disciplines. Second, we evaluate disciplines that show the highest jump from the two-year correlation coefficient to its five-year counterpart. IS ranks numero uno on this list, indicating that IS discipline needs time to match academic impact with practice impact. Third, *American Economic Review*, *Quarterly Journal of Economics*, and *Journal of Marketing Research* occupy top slots on the correlation between two-year IF and two-year attention score. On the correlation between the five-year values, the top three journals are the *American Economic Review*, *California Management Review*, and the *Journal of Business Venturing*. That is, *American Economic Review* stands out as a top journal on both correlations.

Our article contributes by quantifying the match—or the lack thereof—between a journal's and a discipline's impacts on academics and practitioners. In choosing all the journals that *Financial Times* uses in its "Research Rank," ([13]) we offer evidence on the population of journals that business schools use to reward their faculty and news organizations use to rank business schools. We foresee this evidence being useful to multiple stakeholders of academic research. (1) Journal editors can use the evidence to devise ways to boost the translation of academic research into impact on practice. (2) We make a case for business school deans to consider practice impact and not merely citation count (which proxies academic impact) in promoting and tenuring faculty members. (3) Organizations that rate and rank business schools (e.g., Financial Times, U.S. News & World Report) may consider in their evaluation

the practice impact of research produced a school's faculty. (4) We enable manuscript-submitting authors to rank-order journals by their odds of impacting practice and thus make an informed choice of which journal to submit their manuscript. (5) Research-oriented applicants to business programs may benefit from knowing the heterogeneity across disciplines and journals. (6) University and corporate libraries—which serve patrons that look up to academic journals for cutting-edge insights—would benefit from knowing which journals they should carry to boost the value to patrons. Perhaps, the timing is opportune as generative artificial intelligence (AI) chatbots (e.g., OpenAI's ChatGPT, Google's Bard, and Baidu's Ernie) re-emphasize the need for university faculty to *create* knowledge as opposed to merely disseminate knowledge that these chatbots can easily summarize.

## The case for managerial relevance of academic business research

The *Academy of Management Journal* asked practitioners to rate the relevance of academic business research on a scale of 1 (low) to 5 (high). The mean rating was a disappointing 1.11 ([14]). What is more concerning is that respondents ranked academics as the least helpful source for managerial problem-solving ([14]). [15] asked practitioners to evaluate academic journals and reported a correlation of .653. Collectively, these statistics point to the glaring lack of relevance —or should we be bold to claim "irrelevance"—of academic business research.

Because business is an applied discipline, one expects the research produced by business academics to be managerially relevant. *Managerial relevance* Practical relevance is the broader term for managerial relevance ([16]). The former considers nonmanagerial users of academic research, such as society, politicians, regulators, nongovernmental and nonprofit organizations, journalists, lawyers, and social advocates and activists. Academics have used the following terms as related to relevance: importance ([17]), interesting ([18]), impact, influence ([19]), usefulness ([17]), speed ([14]), the "great divide" between academics and practitioners ([20]), and translation ([7]) is "the degree to which a specific manager in an organization perceives academic knowledge to aid his or her job-related thoughts or actions in the pursuit of organizational goals" ([21], p. 212). Unfortunately, academics in almost all business disciplines (e.g., [6, 18, 22–25] have lamented the *lack* of relevance of academic business research. Unfortunately, the conversation has not progressed over the years, with articles published from 1970s ([26]) to 2020s (e.g., [19, 27]) reporting a lack of relevance.

[28] analyzed news and social media citations of 15,900 marketing articles published between 2011 and 2019. The authors reported that 90% (50%) of these articles received zero citation in news (social) media, thus quantifying the lack of relevance of academic marketing research. Relatedly, [29] surveyed professors, associate deans, and external constituents to conclude that business schools assign little weight to relevance of their faculty's research, thus suppressing faculty's incentive to engaging in relevant research. The insight is that the relevance is compromised in the production—and not translation—stage of research ([29]).

Almost all the above commentaries are based on authors' perceptions of lack of relevance (e.g., [23, 24]). Our search of the literature suggests very few articles that have used managerial surveys to measure practice impact (e.g., [30] is a welcome exception). Further, we have found no reports of an objective measure of the impact of journals and articles on practice. Our article fills this gap.

## Data and method

### Selection of journals

We consider the list of journals *Financial Times* (FT) uses for ranking business schools on their research productivity (FT names this rank the *FT research rank*, and we call it the *FT list*, for brevity) [13].

The "first" FT list included 40 journals. In December 2010, FT included the following five journals to its list: *Contemporary Accounting Review*, *Journal of Consumer Psychology*, *Journal of Management Studies*, *Organization Studies*, and *Production and Operations Management*. Thus, the list expanded from 40 to 45 journals https://en.wikipedia.org/wiki/Wikipedia: WikiProject_Academic_Journals/FT_Top_40 and https://personal.utdallas.edu/~mikepeng/ documents/CV201002_TopTierOnly_000.pdf.

On September 12, 2016, FT announced a new list of 50 journals, dropping four journals from its earlier list of 45, and including nine new journals (i.e., 45 − 4 + 9 = 50) [13]. The four journals that FT excluded are *Academy of Management Perspectives*, *California Management Review*, *Journal of the American Statistical Association*, and *RAND Journal of Economics*. The nine journals that FT's new list included are (in alphabetical order): *Human Relations*, *Journal of Management*, *Journal of Management Information Systems*, *Journal of the Academy of Marketing Science*, *Manufacturing and Service Operations Management*, *Research Policy*, *Review of Economic Studies*, *Review of Finance*, and *Strategic Entrepreneurship Journal*.

We started with the FT's revised list of 50 journals and added to the list (1) the **four** journals that FT had excluded in 2016, (2) *Journal of Computing*—the only journal that is not a member of the FT list but features in the University of Texas at Dallas' research ranking of top 100 business schools https://web.archive.org/web/20220307053536/https://jsom.utdallas.edu/the-utd-top-100-business-school-research-rankings/list-of-journals, and (3) **three** other journals: *Journal of Business Research*, *Journal of International Marketing*, and *Marketing Letters*. So, in all, we considered 58 journals.

Two of these 58 journals—*Academy of Management Perspectives* and *Sloan Management Review*—had no data in the Web of Science Journal IF database effective 2002. Therefore, we excluded these journals. At the end of this step, our sample comprised 56 journals.

We classified the 56 journals into 12 disciplines namely, (a) accounting, (b) business ethics, (c) economics, (d) entrepreneurship, (e) finance, (f) general management, (g) information systems, (h) international business, (i) marketing, (j) operations, (k) organizational behavior, and (l) statistics (read Table 1).

## Key measures

Our unit of analysis is journal-year. Consider journal $i$ in year $t$. Our first variable are $IF_{i,t,n}$, where $n$ is either 2 or 5, indicating journal $i$'s two-year IF in year $t$, and its five-year IF in year $t$. Our second variable is *Attention Score*$_{i,t,n}$, where $i$, $t$, and $n$ have the same meanings as in $IF_{i,t,n}$. We describe each of IF and attention score next.

## Impact on academics: IF

Academic journals use *IF* to measure a journal's impact on academics. The Web of Science calculates a journal's IF in intervals of two-year, five-year, and ten-year. Whereas the two-year IF shows the immediate impact of research articles, the five- and ten-year IFs measure the impact in the longer-term. Therefore, we collected from the *Web of Science Journal Citation Reports* data on the two-year and five-year IFs for each journal for each year from 2000 to 2020. Alternatively stated, we did not compute the 10-year IF for two reasons. First, the Web of Science does not provide data for 10-year IF (that is, it provides data for only two- and five-year IF for each journal). Second, the Digital Object Identifier [DOI] became more commonly used effective the year 2000. Consequently, many journals would not have enough values for the 10-year Altmetric attention scores.

**Table 1.** The list of analyzed journals in 12 business disciplines.

| Journal name | Range of Years for *Two*-Year *IF* | Range of Years for *Five*-Year *IF* | Range of Years for *Two*-Year *Attention Score* | Range of Years for *Five*-Year *Attention Score* | Discipline |
|---|---|---|---|---|---|
| *Academy of Management Journal* | 2000–2020 | 2007–2020 | 2001–2022 | 2001–2022 | General management (13) |
| *Academy of Management Perspectives* | 2007–2020 | 2007–2020 | 2012–2022 | 2012–2022 | |
| *Academy of Management Review* | 2000–2020 | 2007–2020 | 2001–2022 | 2001–2022 | |
| *Administrative Science Quarterly* | 2008–2020 | 2008–2020 | 2009–2022 | 2009–2022 | |
| *California Management Review* | 2000–2020 | 2007–2020 | 2001–2022 | 2001–2022 | |
| *Harvard Business Review* | 2000–2020 | 2007–2020 | 2001–2022 | 2001–2022 | |
| *Journal of Management* | 2000–2020 | 2007–2020 | 2001–2022 | 2001–2022 | |
| *Journal of Management Studies* | 2000–2020 | 2007–2020 | 2001–2022 | 2001–2022 | |
| *Strategic Management Journal* | 2000–2020 | 2007–2020 | 2001–2022 | 2001–2022 | |
| *Research Policy* | 2000–2020 | 2007–2020 | 2001–2022 | 2001–2022 | |
| *Organization Studies* | 2001–2020 | 2007–2020 | 2006–2022 | 2006–2022 | |
| *Organization Science* | 2000–2020 | 2007–2020 | 2001–2022 | 2001–2022 | |
| *Management Science* | 2000–2020 | 2007–2020 | 2001–2022 | 2001–2022 | |
| *Journal of Finance* | 2002–2020 | 2007–2020 | 2003–2022 | 2003–2022 | Finance (5) |
| *Journal of Financial and Quantitative Analysis* | 2000–2020 | 2007–2020 | 2001–2022 | 2001–2022 | |
| *Journal of Financial Economics* | 2000–2020 | 2007–2020 | 2001–2022 | 2001–2022 | |
| *Review of Finance* | | | | | |
| *Review of Financial Studies* | 2010–2020 | 2013–2020 | 2001–2022 | 2001–2022 | |
| *Accounting, Organizations & Society* | 2000–2020 | 2007–2020 | 2001–2022 | 2001–2022 | Accounting (6) |
| *Accounting Review* | 2000–2020 | 2007–2020 | 2001–2022 | 2001–2022 | |
| *Contemporary Accounting Research* | 2004–2020 | 2007–2020 | 2001–2022 | 2001–2022 | |
| *Journal of Accounting & Economics* | 2000–2020 | 2007–2020 | 2001–2022 | 2001–2022 | |
| *Journal of Accounting Research* | 2007–2020 | 2007–2020 | 2001–2022 | 2001–2022 | |
| *Review of Accounting Studies* | 2006–2020 | 2008–2020 | 2005–2022 | 2005–2022 | |
| *American Economic Review* | 2000–2020 | 2007–2020 | 2001–2022 | 2001–2022 | Economics (6) |
| *Econometrica* | 2000–2020 | 2007–2020 | 2001–2022 | 2001–2022 | |
| *Journal of Political Economy* | 2000–2020 | 2007–2020 | 2001–2022 | 2001–2022 | |
| *Quarterly Journal of Economics* | 2000–2020 | 2007–2020 | 2001–2022 | 2001–2022 | |
| *RAND Journal of Economics* | 2000–2020 | 2007–2020 | 2001–2022 | 2001–2022 | |
| *Review of Economic Studies* | 2000–2021 | 2006–2021 | 2001–2022 | 2001–2022 | |
| *Journal of Operations Management* | 2000–2020 | 2007–2020 | 2001–2022 | 2001–2022 | Operations (4) |
| *Production and Operations Management* | 2008–2020 | 2008–2020 | 2009–2022 | 2009–2022 | |
| *Operations Research* | 2000–2020 | 2007–2020 | 2001–2022 | 2001–2022 | |
| *Manufacturing & Service Operations Management* | 2000–2020 | 2007–2020 | 2001–2022 | 2001–2022 | |
| *Journal of Applied Psychology* | 2000–2020 | 2007–2020 | 2001–2022 | 2001–2022 | Organizational behavior (4) |
| *Human Relations* | 2002–2020 | 2008–2020 | 2003–2022 | 2003–2022 | |
| *Human Resource Management* | 2009–2020 | 2009–2020 | 2001–2022 | 2001–2022 | |
| *Organizational Behavior and Human Decision Processes* | 2000–2020 | 2007–2020 | 2001–2022 | 2001–2022 | |
| *Journal of Management Information Systems* | 2000–2020 | 2007–2020 | 2001–2022 | 2001–2022 | Information systems (3) |
| *INFORMS Journal on Computing* | 2001–2020 | 2007–2020 | 2001–2022 | 2001–2022 | |

*(Continued)*

**Table 1.** (Continued)

| Journal name | Range of Years for *Two*-Year *IF* | Range of Years for *Five*-Year *IF* | Range of Years for *Two-Year Attention Score* | Range of Years for *Five-Year Attention Score* | Discipline |
|---|---|---|---|---|---|
| *MIS Quarterly* | 2000–2020 | 2007–2020 | 2001–2022 | 2001–2022 | |
| *Journal of the Academy of Marketing Science* | 2000–2020 | 2007–2020 | 2001–2022 | 2001–2022 | Marketing (9) |
| *Journal of Business Research* | 2000–2021 | 2006–2021 | 2001–2022 | 2001–2022 | |
| *Journal of Consumer Psychology* | 2000–2020 | 2007–2020 | 2001–2022 | 2001–2022 | |
| *Journal of Consumer Research* | 2000–2020 | 2007–2020 | 2001–2022 | 2001–2022 | |
| *Journal of International Marketing* | 2004–2021 | 2008–2021 | 2009–2022 | 2009–2022 | |
| *Journal of Marketing* | 2003–2020 | 2007–2020 | 2007–2022 | 2007–2022 | |
| *Journal of Marketing Research* | 2005–2020 | 2007–2020 | 2008–2022 | 2008–2022 | |
| *Marketing Letters* | 2003–2021 | 2007–2021 | 2001–2022 | 2005–2022 | |
| *Marketing Science* | 2000–2020 | 2007–2020 | 2001–2022 | 2001–2022 | |
| *Journal of the American Statistical Association* | 2000–2020 | 2007–2020 | 2001–2022 | 2001–2022 | Statistics (1) |
| *Journal of International Business Studies* | 2000–2020 | 2007–2020 | 2001–2022 | 2001–2022 | International business (1) |
| *Entrepreneurship Theory & Practice* | 2005–2020 | 2008–2020 | 2009–2022 | 2009–2022 | Entrepreneurship (3) |
| *Journal of Business Venturing* | 2000–2020 | 2007–2020 | 2001–2022 | 2001–2022 | |
| *Strategic Entrepreneurship Journal* | 2010–2020 | 2010–2020 | 2008–2022 | 2008–2022 | |
| *Journal of Business Ethics* | 2000–2020 | 2007–2020 | 2001–2022 | 2001–2022 | Business ethics (1) |

## Formula for IF

The formula for the *n*-year IF for a journal in the year *t* is:

$$IF_{t,n} = \frac{times\ articles\ published\ by\ the\ journal\ in\ t-1\ throught\ t-n\ were\ cited\ in\ year\ t}{articles\ published_{t-1} + \ldots articles\ published_{t-n}}$$

For example, one would use the following formula to measure the two-year IF of a journal in the year 2022:

$$IF_{2022,2} = \frac{times\ articles\ published\ by\ the\ journal\ in\ 2020\ and\ 2021\ were\ cited\ in\ 2022}{articles\ published_{2020} + articles\ published_{2021}}$$

## Impact on practitioners: Attention score

We measure a journal's impact on practitioners by Altmetric's *attention score* of each article published in the focal journal-year for which we had IF. For an article, the attention score measures the attention it received from nonacademic stakeholders until the current date, which in our case was March 4, 2022.

The Altmetric attention score is a weighted count of the number of citations the article received across 19 sources until the current date (read Table 2) ([9, 10]).

The score is calculated based on three main factors (Altmetric 2023). First is the **volume** of attention the article receives. The score rises as more people mention the article. Altmetric counts only one mention per source. For example, if the same Twitter account tweets about a research article more than once, Altmetric considers the number of mentions as 1. However, if two accounts tweet about an article, Altmetric considers the two tweets as independent mentions. Second, the *source* of attention determines how each mention is weighted. For example, a news article contributes more to an article's Altmetric attention score than social media posts (such as Twitter or Reddit) ([9]). A higher weight for news sources than social media posts

**Table 2. Sources of Altmetric attention score and weight of each.**

| Source | Weight |
|---|---|
| News | 8 |
| Blog | 5 |
| Policy document (per source) | 3 |
| Patent | 3 |
| Wikipedia | 3 |
| Peer review (Publons, Pubpeer) | 1 |
| Weibo (historical only–since 2015) | 1 |
| Google+ (historical only–since 2019) | 1 |
| F1000 | 1 |
| Syllabi (Open Syllabus) | 1 |
| LinkedIn (historical only–since 2014) | 0.5 |
| Twitter (tweets and retweets) | 0.25 |
| Facebook (curated list of public Pages) | 0.25 |
| Reddit | 0.25 |
| Pinterest (historical only–since 2013) | 0.25 |
| Q&A (Stack Exchange) | 0.25 |
| YouTube | 0.25 |
| Number of Mendeley readers | 0 |
| Number of Dimensions and Web of Science citations | 0 |

asserts that the former bring more attention than the latter. Sources with a weight of zero are not included in the Altmetric score, but are included in the summary overview page on the Altmetric platform. Third, Altmetric determines whether the *author* of a mention is potentially biased. For example, if one account is mentioning the same article multiple times, the mentions are weighted downward. This factor reflects the diversity of individuals discussing the academic article. A combination of these factors produces a weighted approximation of attention a particular research article received until the current date ([9]).

## Calculating two-year and five-year attention scores at the level of journal-year

To calculate the two-year and five-year attention scores for each journal, we used the formula for journal IF. We note one caveat, though. Altmetric only provides cumulative attention data on a per article basis. For example, if an article's attention score was 25 in 2019 but 30 in 2022, users only see the current attention score. Historical data on mention sources is provided but

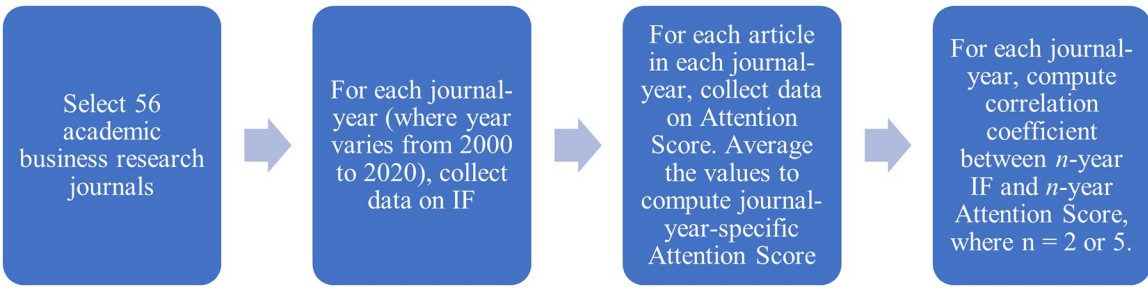

**Fig 1. Research process.** Note: The above figure lists the step-by-step process we adopted.

the proprietary attention score is not historical. Therefore, we use the total attention score on the date (March 4, 2022) Altmetric provided us access to the data. It would be impossible to reconstruct the attention score from scratch at each historical time point because many tweets and other sources are not datetime stamped consistently.

For example, to calculate a two-year attention score for *Econometrica* in the year 2020 we summed the total attention scores for each of the articles published in 2018 and 2019 and divided the sum by the number of articles *Econometrica* published in 2018 and 2019. Fig 1 below presents the research process.

## Results

We aim to measure the correlation between (1) a discipline's impact on academics and practitioners, and (2) a journal's impact on academics and practitioners. We use the Pearson correlation coefficient. For each level of observation—discipline and journal—we use two correlation coefficients to achieve our aim. First is the correlation coefficient between a discipline's (or a journal's) two-year IF and two-year attention score. For ease of exposition, we call this coefficient "two-year correlation." We compute the "five-year correlation" similarly (read Table 3). A low value on either of these two correlations suggests low *balance* between impacting academics and practitioners, whereas a high correlation indicates that the discipline/journal are striking a high balance in impacting both sets of stakeholders: the academics and the practitioners. Fig 2 visualizes the correlation coefficients.

We make three observations from Table 3 and Fig 1. First, international business (IB) leads by having the highest correlation in both two-year and five-year duration (.86 and .93, respectively, and the longest gray and black bars). Entrepreneurship is a close second. IB and entrepreneurship outperforming other disciplines pleasantly surprised us. IS has the weakest correlations, whereas each of accounting and finance ranks low. The weaker performance of these three disciplines (IS, accounting, and finance) surprised us because they are highly applied and empirically driven disciplines as compared to, say, business ethics and general management, which are more theory-based.

Second, IS shows the steepest jump from two-year correlation to its five-year counterpart, with values increasing from .07 to .44. Accounting also shows a sharp increase, albeit less than IS. The insight is that these two disciplines take longer to convert academic impact into practice impact.

**Table 3. Correlation coefficients between two-year (five-year) IF and two-year (five-year) attention score, by the 12 disciplines.**

| Discipline | 2-year correlation | 5-year correlation |
|---|---|---|
| Accounting | 0.43 | 0.75 |
| Business Ethics | 0.68 | 0.83 |
| Economics | 0.55 | 0.56 |
| Entrepreneurship | 0.76 | 0.91 |
| Finance | 0.43 | 0.45 |
| General Management | 0.58 | 0.64 |
| Information Systems | 0.07 | 0.44 |
| International Business | 0.86 | 0.93 |
| Operations | 0.53 | 0.82 |
| Organizational Behavior | 0.62 | 0.79 |
| Marketing | 0.63 | 0.79 |
| Statistics | 0.68 | 0.67 |

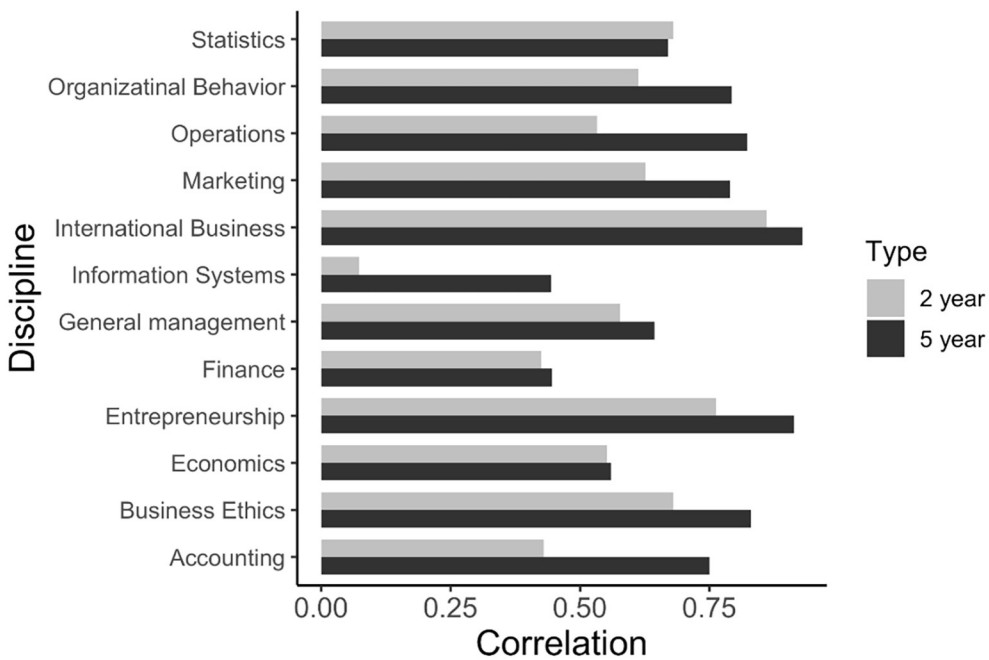

**Fig 2. Correlation coefficients between two-year (five-year) IF and two-year (five-year) attention score, by the 12 disciplines.** Note: The gray bar indicates the correlation coefficient between the two-year IF and the two-year attention score. The black bar reports the coefficient between the five-year values. The insight resides in comparing the difference between the length of the gray bar and the length of the black bar. The higher the difference, the more the time the discipline takes to translate academic impact to practice impact.

Third, except for statistics, no discipline sees a drop from the two-year correlation to the five-year correlation. Even for statistics, the drop is marginal from .68 to .67.

We next repeat the two correlations, albeit for journals and not disciplines. Table 4 lists the correlation.

Fig 3A through 3L provide the bar graphs for each of the 12 disciplines. We make four observations from the results in Table 4 and Fig 2. First, the results show that correlations between journal IF and Altmetric attention score are strong and positive for most journals.

Second, interesting anomalies are the negative correlations (−.56 two-year and −.78 five-year) for *Harvard Business Review* as well as the weaker negative correlations for *Econometrica* (−.11 two-year and −.26 five-year) and *Journal of Financial and Quantitative Analysis* (−.06 two-year and −.10 five-year).

**Table 4. Correlation coefficients between two-year (five-year) IF and two-year (five-year) attention score, by each of 56 *journals*.**

| Journal name | Discipline | Two-Year Correlation | Five-Year Correlation |
|---|---|---|---|
| Academy of Management Journal | General management (14) | 0.78 | 0.80 |
| Academy of Management Perspectives | | 0.77 | 0.62 |
| Academy of Management Review | | 0.72 | 0.81 |
| Administrative Science Quarterly | | 0.61 | 0.82 |
| California Management Review | | 0.78 | 0.97 |
| Harvard Business Review | | -0.56 | -0.78 |
| Journal of Management | | 0.64 | 0.88 |
| Journal of Management Studies | | 0.72 | 0.85 |

*(Continued)*

**Table 4.** (Continued)

| Journal name | Discipline | Two-Year Correlation | Five-Year Correlation |
|---|---|---|---|
| Management Science | | 0.78 | 0.82 |
| Organization Studies | | 0.67 | 0.89 |
| Organization Science | | 0.38 | 0.26 |
| Research Policy | | 0.68 | 0.91 |
| Strategic Management Journal | | 0.53 | 0.52 |
| Journal of Finance | Finance (4) | 0.24 | 0.30 |
| Journal of Financial and Quantitative Analysis | | -0.06 | -0.10 |
| Journal of Financial Economics | | 0.85 | 0.90 |
| Review of Financial Studies | | 0.67 | 0.68 |
| Accounting, Organizations & Society | Accounting (6) | 0.63 | 0.85 |
| Accounting Review | | 0.82 | 0.92 |
| Contemporary Accounting Research | | 0.46 | 0.82 |
| Journal of Accounting & Economics | | -0.12 | 0.15 |
| Journal of Accounting Research | | 0.61 | 0.85 |
| Review of Accounting Studies | | 0.18 | 0.91 |
| American Economic Review | Economics (6) | 0.96 | 0.99 |
| Econometrica | | -0.11 | -0.26 |
| Journal of Political Economy | | 0.73 | 0.71 |
| Quarterly Journal of Economics | | 0.95 | 0.95 |
| RAND Journal of Economics | | 0.16 | 0.23 |
| Review of Economic Studies | | 0.62 | 0.74 |
| Journal of Operations Management | Operations (4) | 0.54 | 0.85 |
| Operations Research | | 0.46 | 0.78 |
| Production and Operations Management | | 0.39 | 0.83 |
| Manufacturing & Service Operations Management | | 0.74 | 0.83 |
| Journal of Applied Psychology | Organizational behavior (4) | 0.85 | 0.87 |
| Human Relations | | 0.71 | 0.87 |
| Human Resource Management | | 0.28 | 0.72 |
| Organizational Behavior and Human Decision Processes | | 0.61 | 0.71 |
| Journal of Computing | Information systems (5) | 0.13 | 0.33 |
| Journal of Management Information Systems | | 0.24 | 0.66 |
| MIS Quarterly | | -0.15 | 0.34 |
| Journal of the Academy of Marketing Science | Marketing (9) | 0.89 | 0.89 |
| Journal of Business Research | | 0.80 | 0.90 |
| Journal of Consumer Psychology | | 0.46 | 0.81 |
| Journal of Consumer Research | | 0.67 | 0.85 |
| Journal of International Marketing | | 0.85 | 0.91 |
| Journal of Marketing | | 0.74 | 0.84 |
| Marketing Letters | | 0.11 | 0.27 |
| Journal of Marketing Research | | 0.95 | 0.95 |
| Marketing Science | | 0.17 | 0.68 |
| Journal of the American Statistical Association | Statistics (1) | 0.68 | 0.67 |
| Journal of International Business Studies | International business (1) | 0.86 | 0.93 |
| Entrepreneurship Theory & Practice | Entrepreneurship (3) | 0.90 | 0.95 |
| Journal of Business Venturing | | 0.93 | 0.96 |
| Strategic Entrepreneurship Journal | | 0.46 | 0.83 |
| Journal of Business Ethics | Business ethics (1) | 0.68 | 0.83 |

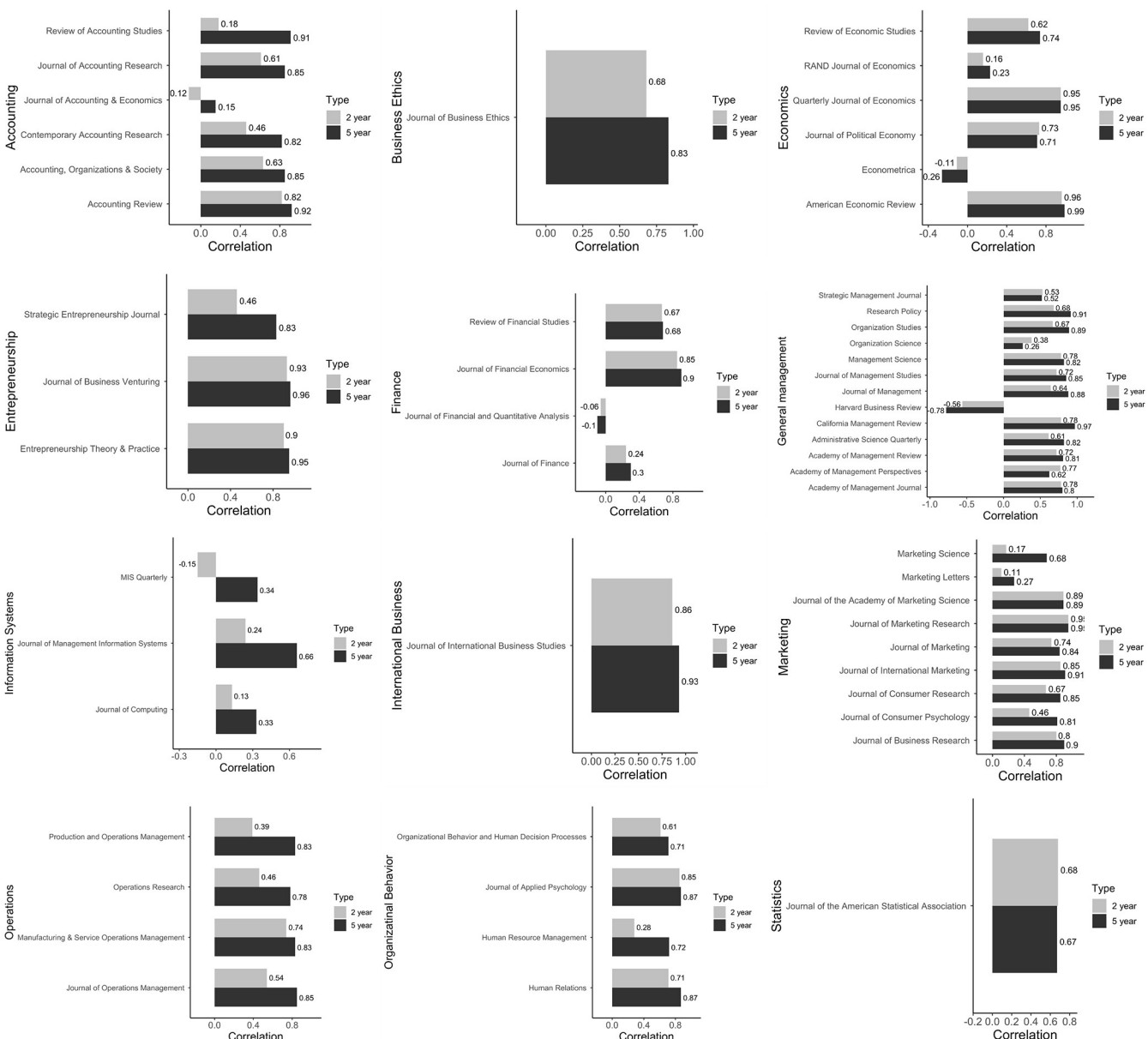

**Fig 3. A. Accounting Journals: Correlation Coefficients between Two-Year (Five-Year) IF and Two-Year (Five-Year) Attention Score.** Note: Comparing a journal's two-year correlation coefficient with the five-year value indicates whether the journal needs more time to translate its academic impact to practice impact. For example, *Review of Accounting Studies* has a low coefficient of two-year correlation (.18) but a high coefficient of five-year correlation (.91), suggesting that the journal tales longer for translation. Conversely, *Accounting Review* does not need much time. **B. Business Ethics Journal: Correlation Coefficients between Two-Year (Five-Year) IF and Two-Year (Five-Year) Attention Score.** Note: The above figure suggests that *Journal of Business Ethics* requires more time to translate it academic impact to practice impact. **C. Economics Journals: Correlation Coefficients between Two-Year (Five-Year) IF and Two-Year (Five-Year) Attention Score.** Note: The above figure suggests that *RAND Journal of Economics* lags in translating academic impact to practice impact, whereas *American Economic Review* aces the translation. **D. Entrepreneurship Journals: Correlation Coefficients between Two-Year (Five-Year) IF and Two-Year (Five-Year) Attention Score.** Note: Each of *Journal of Business Venturing* and *Entrepreneurship Theory & Practice* rates very high on two-year and five-year correlation coefficients. However, *Strategic Entrepreneurship Journal* scores low. **E. Finance Journals: Correlation Coefficients between Two-Year (Five-Year) IF and Two-Year (Five-Year) Attention Score**. Note: *Journal of Finance Economics* is a positive outlier among the four finance journals with a very high two-year and five-year correlation coefficients. *Journal of Finance* has low coefficients, while *Review of Financial Economics* lies in between these extremes. **F. General Management Journals: Correlation Coefficients between Two-Year (Five-Year) IF and Two-Year (Five-Year) Attention Score.** Note: General management journals exhibit high variation in their two-year and five-year correlation coefficients. For example, *Organization Science* has very low coefficients, whereas *Research Policy* and *Organization Studies* have the highest. Overall, most journals hover in the middle values of about [13]. **G: IS Journals: Correlation Coefficients between Two-Year (Five-Year) IF and Two-Year (Five-Year) Attention Score.** Note: All three IS journals have weak coefficients. Further, these journals need time to translate academic impact to practice impact. **H: IB Journal: Correlation Coefficients between Two-Year (Five-Year) IF and Two-Year (Five-Year) Attention Score.** Note: *JIBS* has a high coefficient for both two-year and five-year periods, indicating that the journal does well to

timely translate its academic impact to practice impact. **I: Marketing Journals: Correlation Coefficients between Two-Year (Five-Year) IF and Two-Year (Five-Year) Attention Score.** Note: *Journal of Marketing Research* outperforms all other marketing journals. Further, it enjoys the same coefficient for two-year and five-year periods. *Marketing Letters* sits at the bottom with very weak coefficients. Lastly, *Marketing Science* requires more time to translate its impact. **J: Operations Journals: Correlation Coefficients between Two-Year (Five-Year) IF and Two-Year (Five-Year) Attention Score.** Note: OM journals need time to translate their impact. Further, all four journals achieve high coefficients in five-year period. **K: OB Journals: Correlation Coefficients between Two-Year (Five-Year) IF and Two-Year (Five-Year) Attention Score.** Note: *Human Resource Management* is an outlier with very low value of two-year correlation coefficient. All other three journals perform well on both two-year and five-year coefficients. **L: Statistics Journal: Correlation Coefficients between Two-Year (Five-Year) IF and Two-Year (Five-Year) Attention Score.** Note: *JASA* has moderate but consistent coefficients for two-year and five-year periods.

Third, on average, the five-year correlations tend to be slightly higher than the two-year correlations, except for the *Academy of Management Practices and Journal of Political Economy*. The journal with the largest two-year and five-year differential is the *Review of Accounting Studies*—the difference between its two-year correlation and the five-year correlation is .73.

Fourth, the journals with the largest two-year correlations are the *American Economic Review*, *Quarterly Journal of Economics*, and *Journal of Marketing Research*—which have correlations of .96, .95, and .95. The highest five-year correlations are .99, .97, and .96 for the *American Economic Review*, *California Management Review*, and the *Journal of Business Venturing*, respectively. American Economic Review thus has the highest correlations in both categories.

## Conclusion

Academics in almost all business disciplines have over the years written about the lack of relevance of academic business research. These writings span general management (e.g., [31]), international business (e.g., [4]), marketing (e.g., [5]), information systems (e.g., [6]), and operations (e.g., [24]). Interestingly, none of these article provide quantitative evidence in support of this lack of relevance and usefulness of academic business research.

This article was motivated by this lack of evidence. We took on a massive data collection effort by considering 56 academic business journals from 2000 to 2020 and measuring for each journal-year the two-year IF and two-year attention score, and their five-year counterparts. We computed the correlation coefficients between the two-year values and the five-year values.

A priori, we expected disciplines that are more empirical than theoretical to receive higher correlation coefficients. That is, we expected disciplines of IS, accounting, and finance to perform better than management and international business (IB) disciplines. However, we were pleasantly surprised to see IB and entrepreneurship come on top. We conjecture that more businesspeople read journals in these two disciplines than in other disciplines. We were initially surprised to see that IS, accounting, and finance disciplines as laggards. On second thought, we speculate that the technical language these journals use cause only specialized readers to read the research.

Our evidence offers food for thought for journal editors and business schools' deans and faculty in determining how they can boost the correlations between academic impact and practice impact. Because most business schools are funded by taxpayers, we also call for evidence on the various dimensions of practice impact. For example, Altmetric considers 19 sources of practice impact. We reason that citations of academic research in policy documents, patents, and news are more impactful than citations in social media posts. Deans and journals may consider a system where authors are rewarded disproportionately more for citations of their articles in policy documents and patents than in social media. Such reward system will create appropriate incentives and make business research more accountable.

We also need research that documents impediments to the lack of practice impact of academic research. An intuitive thesis is that academic articles are overly long, technical, backward looking,

and infused with low readability. Future research can analyze practice impact at article level and identify correlations between an article's attention score and its characteristics such as readability, technicality, etc. Future research can also consider using Altmetric Attention Score data to measure the impact of exogenous shocks on a journal's impact on practice. For example, *FT* has included and excluded journals from its list in 2010 and 2016, providing exogenous shocks to the journals that were included/excluded and their peers journals in the discipline.

In summary, we believe our article offers a useful start to highlight the gap between the academic impact and practice impact of academic business journals, while paving the path for future research that can bridge this gap.

## Author Contributions

**Conceptualization:** Vivek Astvansh.

**Data curation:** Ethan Fridmanski.

**Formal analysis:** Ethan Fridmanski.

**Writing – original draft:** Vivek Astvansh.

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
