## [Decision Letter · Decision Letter 0]

23 May 2023

PONE-D-23-12557Academic Business Research: Impact on Academics Versus Impact on PracticePLOS ONE

Dear Dr. Astvansh,

Thank you for submitting your manuscript to PLOS ONE. After careful consideration, we feel that it has merit but does not fully meet PLOS ONE’s publication criteria as it currently stands. Therefore, we invite you to submit a revised version of the manuscript that addresses the points raised during the review process.

We look forward to receiving your revised manuscript.

Kind regards,

Radoslaw Wolniak, full professor

Academic Editor

PLOS ONE

Journal Requirements:

Reviewers' comments:

Reviewer's Responses to Questions

**Comments to the Author**

1. Is the manuscript technically sound, and do the data support the conclusions?

Reviewer #1: Partly

Reviewer #2: Yes

2. Has the statistical analysis been performed appropriately and rigorously? 

Reviewer #1: Yes

Reviewer #2: Yes

3. Have the authors made all data underlying the findings in their manuscript fully available?

Reviewer #1: Yes

Reviewer #2: Yes

4. Is the manuscript presented in an intelligible fashion and written in standard English?

Reviewer #1: Yes

Reviewer #2: Yes

5. Review Comments to the Author

Reviewer #1: Dear Authors,

You must be highly commended for a very good research topic. Actually no one , in my opinion, has researched the practical side of journals. Many scientists write articles for reputable journals and nobody wonders whether the models, methods etc. developed are applied in practice. Does the IF journal have an impact on the practical usefulness of the research results presented in journals?

I have done a lot of reviewing already, but this is the first time I am reviewing research results about the pratical impact of journals. Soper, my congratulations to the authors for an interesting research topic. However, I have a few comments:

1: The astract and keywords need to be expanded. In the abstract, the authors should add the purpose of the research, briefly describe the methodology and add the results, please do not limit yourself to only one top journal: American Economic Review, which, as the authors state, " leads all 56". It should be pointed out that there were 12 business disciplines (those in Figure 2) and more about the results. The following should be added or changed to the keywords: journal impact on practices, journal analysis, business disciplines

2: Introduce: the line "First, we use the Web of Science Journal Impact Factor IF", please add the abbreviation and in the rest of the text use only the abbreviation: IF.

After the sentence: We use these scores to compute for each journal-year the two-year and five-year attention score. Third, for each article and each discipline, we compute the correlation coefficient between (1) the two-year impact factor and two-year attention score, and (2) the five-year impact factor and the five-year attention score. The correlation coefficients report three insights.

The hypothesis of the study and the purpose of the study need to be added.

3: Data and methods, in my description you would need to make a drawing - a step by step diagram of the research. A description is good but a drawing always shows the step-by-step nature of the research better. The formula for correlation coefficients (r) should be added.

4: Key Measures

Tabel 1 56 Journals, Belonging to 12 Business Disciplines (without the cut in front of belong) or change title: The list of analysed journals in 12 business disiplines

5: Formula (1 and 2) for Impact Factor . I would use abbreviations and sub-templates to describe them either before the formulae: Impact Factor -IF, time-t, article published in journal - JP (journal paper) etc. Then the formulas would be shorter.

6: Results: note for Fig. 1 has to be moved under the figure. Under each figure, starting with Figure 2A.....2L you have to write one or two sentences of the conclusion, which is shown in the figure. To each business disciplines because this is the core of the research and requires more analysis.

7 Conclusion, too short, I would add: conclusion of the research and then concusion. You need to refer to the hypothesis of the research and the purpose of the research, which needs to be added to the introduction first.

I wonder if the result of the research should not be a ranking of journals.

Please take my comments very kindly, I am impressed with the research and it needs to be presented well.

Regards:

Reviewer

Reviewer #2: A manuscript with a very interesting title. However, the manuscript requires solid refinement, especially from the literature review side. The introduction recommends highlighting what is new at work. The strength is the methodology; the weakness is the very poor review of the literature. The conclusion should also be greatly expanded.

6. PLOS authors have the option to publish the peer review history of their article (what does this mean?). If published, this will include your full peer review and any attached files.

Reviewer #1: No

Reviewer #2: No

---

## [Author Response · Author response to Decision Letter 0]

24 May 2023

Please refer to "Response to Reviewers.docx."

---

## [Decision Letter · Decision Letter 1]

28 Jun 2023

PONE-D-23-12557R1Academic Business Research: Impact on Academics Versus Impact on PracticePLOS ONE

Dear Dr. Astvansh,

Thank you for submitting your manuscript to PLOS ONE. After careful consideration, we feel that it has merit but does not fully meet PLOS ONE’s publication criteria as it currently stands. Therefore, we invite you to submit a revised version of the manuscript that addresses the points raised during the review process.

We look forward to receiving your revised manuscript.

Kind regards,

Radoslaw Wolniak, full professor

Academic Editor

PLOS ONE

Journal Requirements:

Reviewers' comments:

Reviewer's Responses to Questions

**Comments to the Author**

1. If the authors have adequately addressed your comments raised in a previous round of review and you feel that this manuscript is now acceptable for publication, you may indicate that here to bypass the “Comments to the Author” section, enter your conflict of interest statement in the “Confidential to Editor” section, and submit your "Accept" recommendation.

Reviewer #1: (No Response)

Reviewer #2: (No Response)

2. Is the manuscript technically sound, and do the data support the conclusions?

Reviewer #1: Yes

Reviewer #2: Yes

3. Has the statistical analysis been performed appropriately and rigorously? 

Reviewer #1: Yes

Reviewer #2: Yes

4. Have the authors made all data underlying the findings in their manuscript fully available?

Reviewer #1: Yes

Reviewer #2: Yes

5. Is the manuscript presented in an intelligible fashion and written in standard English?

Reviewer #1: Yes

Reviewer #2: Yes

6. Review Comments to the Author

Reviewer #1: (No Response)

Reviewer #2: The manuscript looks much better. However, the review of the literature is still very modest. I recommend that the authors add more literature in 2022 and 2023. Other elements of the work have been improved.

7. PLOS authors have the option to publish the peer review history of their article (what does this mean?). If published, this will include your full peer review and any attached files.

Reviewer #1: No

Reviewer #2: No

---

## [Author Response · Author response to Decision Letter 1]

30 Jun 2023

Please see attached Response to Reviewers.docx.

---

## [Editor Report · Decision Letter 2]

10 Jul 2023

Academic Business Research: Impact on Academics Versus Impact on Practice

PONE-D-23-12557R2

Dear Dr. Astvansh,

We’re pleased to inform you that your manuscript has been judged scientifically suitable for publication and will be formally accepted for publication once it meets all outstanding technical requirements.

Kind regards,

Radoslaw Wolniak, full professor

Academic Editor

PLOS ONE
---

## [Editor Report · Acceptance letter]

20 Jul 2023

PONE-D-23-12557R2 

Academic Business Research: Impact on Academics Versus Impact on Practice 

Dear Dr. Astvansh:

I'm pleased to inform you that your manuscript has been deemed suitable for publication in PLOS ONE. Congratulations! Your manuscript is now with our production department. 

Kind regards, 

on behalf of

Professor Radoslaw Wolniak 

Academic Editor

PLOS ONE